# PLHF: Prompt Learning from Few-shot Human Feedback

## Abstract

Recent advances explore prompt tuning for large language models (LLMs) and develop automatic optimization frameworks to obtain suitable prompts with respect to desired output quality metrics. Although existing approaches can handle conventional tasks such as fixed-solution question answering, defining the metric becomes complicated when the output quality cannot be easily assessed by comparisons with standard golden samples, especially for those natural language applications that multiple outputs are equally valid. Consequently, optimizing the prompts effectively and efficiently without a clear metric becomes a critical challenge. To address this issue, we present PLHF, a few-shot prompt optimization framework inspired by the well-known RLHF technique. Different from näive strategies involving human experts, PLHF employs a specific evaluator module acting as the metric to estimate the output quality. PLHF requires only a single round of human feedback to complete the entire prompt optimization process. Empirical results on both public and industrial datasets show that PLHF significantly outperforms existing output scoring strategies for LLM prompt optimizations.

## 1 Introduction

General-purpose large language models (LLMs) have demonstrated substantial capabilities across various fields in recent years. However, solving complex tasks with LLMs often requires appropriate customizations on LLMs to fit the task requirements. While fine-tuning pre-trained LLMs is a common approach, it may be infeasible when there is limited training data, restricted computational resource, or when working with a black-box LLM. Alternatively, previous studies (Wang et al., 2022; Shin et al., 2020) have shown that the potential of LLMs can also be fully leveraged with suitable *prompts*. Recent literature develops automatic few-shot prompt optimization for LLM usages, such as DSPy (Khattab et al., 2024) and TextGrad (Yuksekgonul et al., 2024). To determine an effective prompt for the LLM, existing methods often employ gradient descent or other algorithms (Yang et al., 2024; Zhou et al., 2023; Guo et al., 2023) to optimize the performance with respect to desired metrics (i.e., definition of output quality). Overall, the key to success heavily relies on the output quality evaluations which shall precisely reveal the model performance to the optimizer.

Although such output quality metrics are often well-defined for the tasks which can be modeled as the traditional discriminative tasks (e.g., classifications and regressions) where the performance can be directly evaluated given ground-truths, scoring the outputs often becomes non-trivial for most generation-type of tasks. An example comes from the essay writing task where the LLM needs to output an essay given specific requirements. To perform prompt optimizations, the evaluator needs to score an essay written by an LLM. This could be extremely challenging without human's involvement. Another typical scenario is the dialogue systems (e.g., chat-bots), as automatically rating the outputs is difficult. In this case, the output quality could be affected by numerous factors such as the context, the circumstances of the environment, as well as specific user preferences. Therefore, a generalized evaluation metric is difficult to formulate.

The absence of a precise metric would hinder the effectiveness of the prompt optimization process for a generative task. Most prompt optimization systems adopt two types of mechanisms to score the generated outputs. The first is employing simple evaluators, such as exact matching or soft matching (based on certain similarity measurement) to compare the generated outputs with the observed samples. The second common strategy is to utilize existing LLMs for the output scoring. For instance,

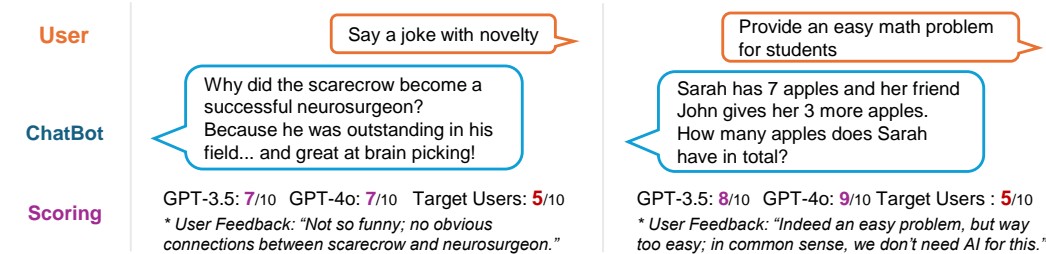

Figure 1: Demonstrations of the actual failure cases that the evaluations from pre-trained LLMs have different preference from specific humans. The first (left) example is the task of joke generation, where the scoring is according to funniness and novelty. The second scenario is a math problem generation bot, where the response quality is evaluated based on helpfulness and problem quality. As shown above, the verdicts of state-of-the-art LLMs could still differ from real human's preferences.

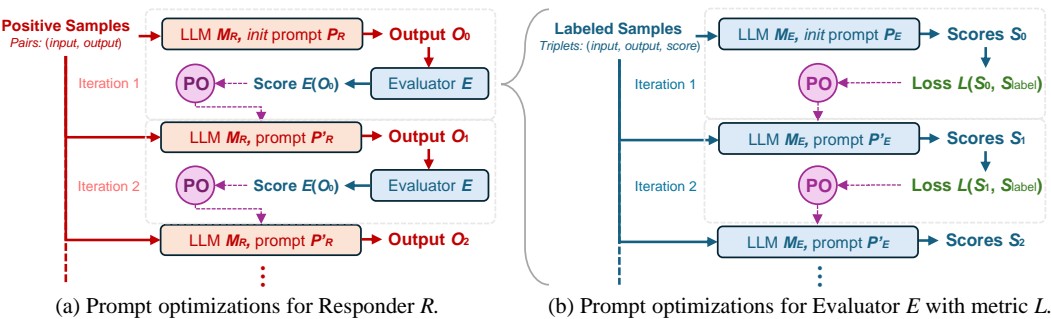

(a) Prompt optimizations for Responder $R$.  (b) Prompt optimizations for Evaluator $E$ with metric $L$.

Figure 2: Workflow framework of PLHF. The entire LLM program contains two modules, Responder $R$ and Evaluator $E$, where PO can be PO arbitrary prompt optimization method.

several studies (Wang et al., 2023; Fu et al., 2024) leverage advanced OpenAI GPT models (Achiam et al., 2023) as an external judge to score the generated outputs. Nevertheless, such scorers suffer from a critical drawback as generic pre-trained LLMs might not have enough contextual or background knowledge to behave as accurate scoring functions. (See Figure 1 for the examples of failure cases.) Ultimately, the purpose of prompt optimization is to tackle complex tasks that the original LLM, when provided with a simple prompt, fails to handle effectively. Consequently, employing such a sub-optimal LLM as the scoring mechanism is likely to diminish the effectiveness of the prompt optimization framework. As a result, for applications related to response generation, it is highly demanded to involve real human experts to evaluate the results generated by LLMs, yet such scheme suffers from a frequently occurring issue — constrained budget to employ human experts.

For most of the automatic prompt optimization frameworks (Khattab et al., 2024; Yuksekgonul et al., 2024; Pryzant et al., 2023; Deng et al., 2022; Wen et al., 2024), multiple iterations of optimizations are performed, with the quality estimated based on the given metric. With querying human experts acting as the metric, once the prompt is updated with any modifications, we have to ask human experts for their judgement again for each of the training inputs, which might cause serious efficiency bottleneck. To address the aforementioned issues, we present PLHF (which stands for **P**rompt **L**earning with **H**uman **F**eedback), a few-shot prompt optimization framework. Inspired by the famous Reinforcement Learning from Human Feedback (RLHF) technique (Ouyang et al., 2022; Li et al., 2023), PLHF introduces a particular *evaluator* module $E$ which requires human scoring no greater than *linear* (with respect to the number of training samples) times during the optimization process. To leverage human feedback in few shots, we consider utilizing a prompt-optimized LLM as $E$ to evaluate the output of the main responder $R$. The overall framework is depicted in Figure 2. Specifically speaking, first, we employ human experts to provide judgements as scores on a set of training samples $D$, containing pairs of the inputs and sample text-outputs. Then, we perform

prompt optimization on another LLM to mimic the human experts' preference pattern based on $D$. Since the prompting task on the evaluator module is relatively typical (e.g., binary classifications or regressions), we can leverage any existing automatic few-shot prompt optimization frameworks (e.g., DSPy (Khattab et al., 2024)) with trivial metrics (e.g., Accuracy or Mean Absolute Error) to obtain the evaluator module. Finally, we can perform prompt optimizations for both $E$ and $R$ to establish the entire framework.

To verify the actual effectiveness of PLHF, experiments are conducted not only on multiple benchmark datasets but also on a real-world industrial data collected from an online customer support chat-bot product of an AI company. The experimental results shown in Section 4 demonstrate that PLHF can boost the output quality of existing automatic few-shot prompt optimization frameworks with our duo-module design. Moreover, to verify the effectiveness of each module, we also provide an analysis towards the relations in performance curves between the number of training samples and the output scores for each subtask. The source codes for the experiments are available at https://(*keeping secret for the double-blind paper review*).

In summary, our contributions are as follows:

- We study automatic few-shot prompt optimization for LLMs with limited number of human feedback calls, which is a more reasonable and feasible setting in real-world applications, especially for those have specific user preferences or multiple acceptable outputs.
- We introduce PLHF, a novel prompt optimization framework that does not directly rely on well-defined metrics for the text output. Instead, we design an evaluator module to provide an automatic mechanism that evaluates text outputs for LLM program prompting.
- We conduct extensive experiments on publicly accessible benchmark datasets as well as a test on industrial data to validate the effectiveness of PLHF. The results show that PLHF has superiority in terms of output quality, compared with the approaches employing string matching or adopting the state-of-the-art LLM (GPT-4o) as the evaluator.

## 2 RELATED WORK

Recent research has investigated various strategies to obtain suitable prompts for LLMs. Earlier studies have introduced techniques of automating the search process for data samples (Gao et al., 2021), learning prompts through gradient-based searching methods (Shin et al., 2020; Wen et al., 2024; Pryzant et al., 2023), refining prompts using evolutionary algorithms (Guo et al., 2023; Fernando et al., 2023) and utilizing other LLMs for prompt generation (Yang et al., 2024; Zhou et al., 2023). Several studies have also attempted to optimize prompts using reinforcement learning, exploring prompt editing at different granular levels such as word-level (Deng et al., 2022), phrase-level (Zhang et al., 2023), and within text-to-image generation tasks (Hao et al., 2024).

As LLMs are increasingly applied in real-world scenarios, in-context learning (McCann et al., 2018; Radford et al., 2018; Brown et al., 2020) is becoming an emerging trend for effective LLM programming. Instruction tuning (Ouyang et al., 2022) further enhances this process by enabling complex behaviors through the use of structured prompts (Press et al., 2023; Yao et al., 2023; Khot et al., 2023; Madaan et al., 2024).

For automatic few-shot prompt optimization, Khattab et al. (2024) introduced DSPy, a state-of-the-art prompt optimization framework, which considers LLM usages in a programmatic fashion. DSPy parameterizes each module to learn the data pattern and the desired behaviors by iteratively bootstrapping useful demonstrations. On the other hand, Yuksekgonul et al. (2024) inspired by LLM fine-tuning procedures and proposed TextGrad, a framework refining the prompt with the back-propagation algorithm. Instead of deriving numeral-valued gradients, TextGrad regards LLMs' text feedback as the 'gradient' in texts.

All these techniques rely on well-defined metrics to set their objectives. While advanced general-purpose LLMs like GPT-4 can be adopted for text-output evaluations (Zimbres, 2024; Zheng et al., 2023; Tan et al., 2024), they may lack the contextual or background knowledge needed for accurate evaluation in specific tasks. Hence, involving human feedback becomes inevitable for the prompt optimizations in such tasks. To address the issue, Lin et al. (2024) inspired by dueling bandits and designed a strategy to choose pairs of prompts to query for human feedback during the prompt

**Input**: "Why did the jellyfish refuse the job offer?"
**Output**: "Because it couldn't handle the pressure!"
**Labeled Score**: 6

**Input**: "Why did the peacock sit on the fence all day?"
**Output**: "It had some serious trust issues with its own reflection!"
**Labeled Score**: 5

**Input**: "Why did the vegetarian vampire only drink tomato juice?"
**Output**: "Because he couldn't stand steaks through the heart!"
**Labeled Score**: 7

**Input**: "Why did the shoe store owner become a detective?"
**Output**: "Because he was good at solving sneaker cases!"
**Labeled Score**: 9

**Input**: "Why don't astronauts throw parties in space?"
**Output**: "Because there's no atmosphere for a good vibe!"
**Labeled Score**: 8

**Input**: "Why did the smartphone go to therapy?"
**Output**: "It had too many hang-ups!"
**Labeled Score**: 8

**Positive Samples**

**Training Samples $D$ (Labeled Samples)**

**Predict** the Labeled Score for the given Input-Output pair.
Input is the "setup" and Output is the "punchline" of a joke.
---
Follow the following format.
Input: ${input}
Output: ${output}
Labeled Score: ${score}        **Initial Prompt $P_E$ (Instruction)**
---
Input: "Why did the shoe store owner become a detective?"
Output: "Because he was always good at solving sneaker cases!"
Labeled Score: 9

Input: "Why did the jellyfish refuse the job offer?"
Output: "Because it couldn't handle the pressure!"
Labeled Score: 6

**Optimized Prompt $P'_E$**

**Generate** the Output for the given Input, where Input is the "setup" and Output is the "punchline" of a joke.
---
Follow the following format.
Input: ${input}
Output: ${output}        **Initial Prompt $P_R$ (Instruction)**
---
Input: "Why did the shoe store owner become a detective?"
Output: "Because he was always good at solving sneaker cases!"

Input: "Why don't astronauts throw parties in space?"
Output: "Because there's no atmosphere for a good vibe!"

Input: "Why did the smartphone go to therapy?"
Output: "It had too many hang-ups!"

**Optimized Prompt $P'_R$**

Figure 3: A toy example to illustrate the subtask designs of PLHF. The targeted generative AI task for this example is "*generate the punchline for a joke setup*." The training samples $D$ are triplets (*Input*, *Output*, *Labeled Score*), where *Input* is the joke setup, *Output* is a sample output of the punchline for the corresponding *Input*, and *Labeled Score* is the rating judged by human experts. For this example, we consider *Labeled Score* $\geq 8$ as the condition of positive samples. Examples of optimized prompts $P'_E$ and $P'_R$ (for the evaluator $E$ and the responder $R$, respectively) are shown.

optimizations to reduce the number of needed calls of human feedback. In this paper, we consider a different approach to tackle the issue — we focus on the *metric* in the prompt learning process. We developed a duo-module framework to obtain an evaluator module acting as the metric of the main task to perform the desired LLM prompt optimizations, requiring minimal human feedback.

## 3   PLHF: PROMPT LEARNING FROM FEW-SHOT HUMAN FEEDBACK

As shown in Figure 2, our entire framework, PLHF, is designed to perform prompt optimizations for typical language model program usages — output a proper response based on the given input. The whole process is guided by the principal intuition of taking advantages from few-shot in-context learning Brown et al. (2020) to capture the contextual patterns from limited number of labeled samples. Since there is no explicit metric available, a scoring function is needed for existing prompt optimization frameworks. Hence, we introduce an *evaluator* module $E$, acting as the scoring function for the main *responder* module $R$. The two modules correspond to two respective subtasks.

### 3.1   RESPONDER TASK

The responder task for $R$ is tasked with performing the original assignment. The core of this component is based on a base LLM, denoted as $M_R$, which generates outputs based on the input query. LLM $M_R$ is starting from pairing with an initial prompt $P_R$ describing the roles of input and output, as well as the relationship between input and output (i.e., how the input determines the output). See Figure 3 for an example. The training samples for $R$ include input-output pairs with *positive*

---

**Algorithm 1** PLHF: Duo-module Framework for Few-shot LLM Prompt Optimizations

---

**Input:** Training data samples $D = \{d_1, d_2, \ldots, d_n\}$ (each $d_i$ is a triplet containing input/query $q_i$, output $o_i$ and score/verdict $r_i$), Base LLMs $M_R$, $M_E$, Initial prompts $P_R$, $P_E$, Trivial metric $L$ for evaluator $E$.

**Output:** Optimized prompts $P'_R$ and $P'_E$.

1: Set $P'_R := P_R$ and $P'_E := P_E$.
2: **while** there are new training samples in $D$ **do**
3:     **while** $P'_E$ is not yet optimized (converged) for $D$ **do**
4:         **for** each training sample $d_i = (q_i, o_i, r_i) \in D$ **do**
5:             Input $(q_i, o_i)$ pair to generate the score (or verdict) $\tilde{r}_i$ by $M_E$ with prompt $P'_E$.
6:         **end for**
7:         Compute conventional metric score $S_E = L(\{\tilde{r}_1, \ldots, \tilde{r}_n\}, \{r_1, \ldots, r_n\})$.
8:         Consider the $D$, $P'_E$ and $S_E$ to update the prompt $P'_E$.
9:     **end while**
10:     **while** $P'_R$ is not yet optimized (converged) for $D$ **do**
11:        **for** each training sample $d_i = (q_i, o_i, r_i) \in D$ with *positive* rating $r_i$ **do**
12:           Input $q_i$ to generate the output $\tilde{o}_i$ by $M_R$ with prompt $P'_R$.
13:        **end for**
14:        Evaluate the outputs by evaluator $E$. Obtain the score $S_R = M_E(\{\tilde{o}_1, \ldots, \tilde{o}_n\}; P'_E)$.
15:        Consider the $D$, $P'_R$ and $S_R$ to update the prompt $P'_R$.
16:     **end while**
17:     **if** (*optional*) having new inputs/queries $\{t_1, \ldots, t_m\}$ to augment $D$ **then**
18:        Test the current responder $R = (M_R; P'_R)$ with inputs/queries $\{t_1, \ldots, t_m\}$.
19:        Collect human feedback $f_i$ for the output $M_R(t_i; P'_R)$, for $i = 1, \ldots, m$.
20:        Append $(t_i, M_R(t_i; P'_R), f_i)$, for $i = 1, \ldots, m$, as new training samples into $D$.
21:     **end if**
22: **end while**
23: **return** $P'_R, P'_E$

---

score/verdict labeled by human experts. The data positivity can be specifically defined to align with the requirements of the assigned AI task. For instance, the example described in Figure 3 considers a score threshold as the condition of the positive samples. The generated responses by the LLM $M_R$ with prompt $P_R$ are then judged by the evaluator module $E$. To enhance the quality and adaptability of the responses, with respect to human experts' preference patterns, we perform prompt optimizations on LLM $M_R$ to obtain a *prompt-optimized* prompt $P'_R$ for $M_R$. Finally, we consider $M_R$ with prompt $P'_R$ as the finalized responder module $R$ in our framework to produce desired outputs that solve the original assigned AI task.

### 3.2 Evaluator Task

The evaluator task for $E$ is an auxiliary task designed to verify and score the output generated by the responder $R$. Similar to the responder $R$, the evaluator $E$ is also built with a base LLM, denoted as $M_E$. The evaluator task considers training samples in the triplet form of input, output, and the corresponding score (e.g., verdict, rating) labeled by human experts. Note that, different from the training samples for $R$, this time we consider *all* training data samples (regardless of the positivity) as the references for $E$. To provide a nuanced verdict, we also optimize the prompt $P_E$ for $M_E$ with a trivial metric $L$ (such as the conventional Accuracy or Mean Absolute Error) to evaluate the quality of the response. The metric $L$ is specifically defined as the loss to estimate the difference between the predicted scores and the actual labeled scores in $D$. Overall speaking, The evaluator module $E$ leverages LLM $M_E$ with the finalized prompt $P'_E$ to provide a score rating toward any responses for the original AI task.

With the evaluator $E$, PLHF ensures that the outputs from the responder $R$ are not only technically accurate but also contextually appropriate for the task. The feedback loop between the evaluator $E$ and the responder $R$ helps refine the overall model performance, as the scores determined by $E$ are used to inform future responses generated by $R$.

## 3.3 INTEGRATION AND FEEDBACK LOOPS

With the integration of responder $R$ and evaluator $E$, the entire system operates in a feedback-loop structure, as described in pseudo codes stated in Algorithm 1. At the beginning, we initialize prompts as $P'_R := P_R$ and $P'_E := P_E$ for modules $R$ and $E$, respectively. In each iteration of PLHF, first, training data samples $D$ are used to optimize the evaluator $E$ (i.e., to update $P'_E$). Then, we optimize the responder $R$ (i.e., update $P'_R$) regarding the evaluator $E$ with prompt $P'_E$ as the metric. After an iteration of prompt optimizations for $P'_R$ and $P'_E$, we obtain a version of optimized prompts for the responder $R$ and the evaluator $E$. Figure 3 provides an example of a toy generative AI problem to demonstrate the optimized prompts $P'_R$ and $P'_E$.

For batch tests, the whole optimization process is ended by Line 16 in Algorithm 1 and the finalized prompt $P_R$ are used for the LLM $M_R$ to generate the outputs for upcoming test inputs/queries. On the other hand, if the scenario has incrementing data samples, starting from Line 17 in Algorithm 1, we can augment the training sample set $D$ with the user feedback on new samples, then repeat the optimization processes for $E$ and $R$.

Overall, the proposed PLHF framework is capable of performing prompt optimizations for LLMs even when occurring challenges of (a) no available well-defined metrics to evaluate the LLM output quality for the specific task, (b) limited number (few-shot) of labeled samples for LLM prompting, and (c) multiple valid outputs for a single input.

## 4 EXPERIMENTS

To evaluate the performance and robustness of our proposed model framework, PLHF, we conducted a series of experiments across various tasks. The tasks were selected to test the model ability to generate accurate and contextually relevant outputs, while also assessing the effectiveness of the auxiliary evaluator task in refining responses. In the following subsections, we detail the dataset selection, experimental setup, and the results obtained from these experiments.

### 4.1 DATASETS

We conduct the experiments on three public datasets with various tasks, and one industrial dataset from a real-world product of question answering chat-bot generating practical SQL commands.

#### 4.1.1 SCHEMA GUIDED DIALOGUE (SGD) DATASET

The Schema Guided Dialogue (SGD) dataset Sun et al. (2021) is a large-scale dataset designed for task-oriented dialogue systems. It comprises dialogues collected in English, specifically designed to encompass a wide range of dialogue scenarios, schema-based actions, and services. The dataset contains 1,000 dialogues, contributing to a total of 13,833 utterances. Each user utterance in the dataset is labeled with a *satisfaction score* on a 5-point Likert scale (Likert, 1932). Rating 1 indicates the lowest level of satisfaction, while rating 5 denotes the highest satisfaction. The distribution of satisfaction ratings $\{1, 2, 3, 4, 5\}$ is $\{120, 769, 11151, 1494, 50\}$. These human-assigned satisfaction scores are valuable for assessing chat-bot responses with respect to user satisfaction.

#### 4.1.2 AUTOMATED ESSAY SCORING (AES) DATASETS

The dataset is originally provided by Ben et al. (2012) for the Automated Student Assessment Prize (ASAP). The dataset, named as AES-ASAP, consists of eight essay sets varying in length, ranging from an average of 150 to 550 words per response. The responses were written by students in grades seven through ten, and all essays were hand-graded by human experts. Each essay was double-scored, with a resolved score provided to harmonize the differences between raters. We use the training set (with the scores in domain 1) of the first set of essays in our experiments. The actual text of the student's response is included. We consider the *average score* as aggregated result from these raters. The scores are distributed from 1 to 30. In our experiments, we discard the essays with transcription errors (marked as "illegible" or containing placeholder text such as "???") from the training data.

Apart from AES-ASAP, our experiments also include a newer essay scoring dataset of from an online competition hosted by Kaggle (2024). The dataset, named as AES-2.0, contains 24,000 student-written argumentative essays. Each essay was scored on a scale of 1 to 6 as the holistic rating [1] judged by human experts. Similar to our settings for AES-ASAP, we consider the provided training split for prompt optimizations in our the experiments.

### 4.1.3 INDUSTRIAL SQL COMMAND QUESTION ANSWERING DATASET

In addition to the previously mentioned public datasets, we have also deployed PLHF on a real-world question-answering system, which is currently an actual product of a commercial AI company. This test, named as SQL-QA, comprises 100 real-world queries involving various database inquiry requests from the clients. The database entries contains daily transaction records and other logs sourced from multiple banks in China. As the training data for prompt optimizations, human experts from the company labeled 10 positive samples and 20 negative samples for the prompt optimizations. Table 1 provides examples of queries used in this test.

Table 1: Examples of the queries in our industrial SQL-QA test.

| User Query (English translation) | SQL Statement (Output) |
|---|---|
| Please list the top 3 clients by total deposits at the Beijing branch as of January 31, 2024. | `SELECT CUST_ID, CUST_NAME, DEPO_BAL` 
 `FROM acct` 
 `WHERE DATA_DT='20240131' AND ORG_NAME='Beijing'` 
 `ORDER BY DEPO_BAL DESC` 
 `LIMIT 3` |
| Please inquire about the top 5 banks with the highest asset balances, grouped by institution, as of March 31, 2024. | `SELECT ORG_NAME, ASSET_BAL` 
 `FROM acct` 
 `WHERE DATA_DT='20240331'` 
 `ORDER BY ASSET_BAL DESC` 
 `LIMIT 5` |

### 4.2 EXPERIMENT SETUPS

To perform prompt optimizations (denoted as **PO**) in each subtask, we consider two state-of-the-art automatic prompting frameworks, DSPy and TextGrad. Same experiments are conducted for both frameworks, and respective results are shown.

For the experiments, first, we estimate the effectiveness of the evaluator $E$, which solves the task of predicting the labeled scores based on each input-output pair. The comparisons include several baseline methods:

- **Base LLM (GPT-3.5)**: the grounding baseline — simply employing raw gpt-3.5-turbo-0125 model to predict the labeled score each input-output pair. We utilize OpenAI APIs for the model implementation. The prompt is set to be the same as the initial prompt $P_E$ in PLHF for the comparisons. See Figure 3 for an example.

- **MLP with Text Embedding**: DNN-based predictor — leveraging a Multi-layer Perception (MLP) model, based on the algorithm presented by Popescu et al. (2009), to predict the score of each input-output pair. Since the input and the output are texts, we transform the texts into embeddings by the powerful text-embedding-ada-002 model (OpenAI, 2024) to obtain the respective numerical vectors. The number of layers is set to 3.

- **SVM with Text Embedding**: conventional ML predictor — similar to the MLP one, but this time considering Support Vector Machines (SVMs) as the score predictor. We adopt the SVM implementation provided by Chang & Lin (2011) with the default configuration.

---

[1] https://storage.googleapis.com/kaggle-forum-message-attachments/2733927/20538/Rubric_HolisticEssayScoring.pdf

Table 2: Summary of experimental results for the evaluator subtask across each dataset. For the public datasets, the presented values are RMSE losses (lower is better) of the output scores from $E$; for the industrial dataset SQL-QA, the values indicate Accuracy scores (higher is better). The best ones are marked in bold font.

| Method | SGD | AES-ASAP | AES-2.0 | SQL-QA |
|---|---|---|---|---|
| Base LLM (GPT-3.5) | 1.02 | 4.75 | 0.46 | 0.53 |
| MLP with Text Embedding | 1.17 | 7.22 | 1.08 | 0.33 |
| SVM with Text Embedding | 1.25 | 6.43 | 1.10 | 0.40 |
| Base LLM PO via DSPy | 0.43 | **2.36** | **0.33** | **0.80** |
| Base LLM PO via TextGrad | **0.40** | 2.42 | 0.38 | 0.73 |

- **GPT-3.5 PO via DSPy**: LLM with demonstration-based PO — performing prompt optimizations with DSPy framework (Khattab et al., 2024), based on the aforementioned setting of Base LLM (GPT-3.5).

- **GPT-3.5 PO via TextGrad**: LLM with text instruction-based PO — performing prompt optimizations with TextGrad framework (Yuksekgonul et al., 2024), based on the aforementioned setting of Base LLM (GPT-3.5).

Then, for the main responder task $R$, output quality of the LLM with optimized prompts are judged by test queries. We consider various types of evaluators for the prompt optimizations.

- **Base LLM (GPT-3.5)**: the grounding baseline — simply utilizing raw gpt-3.5-turbo-0125 model via OpenAI APIs to generate the output based on the given input. The prompt is set to be the same as the initial prompt $P_R$ in PLHF for the experiments. See Figure 3 for an example.

- **PO with GPT-4o**: using a state-of-the-art LLM as the evaluator — performing PO on the Base LLM (GPT-3.5). Adopting GPT-4o (gpt-4o-2024-05-13) model as the judge to score the outputs during PO. We employ OpenAI APIs in the implementation.

- **PO with Exact Matching**: scoring by hard-matching —- Base LLM (GPT-3.5) with PO regarding the given reference outputs as the ground-truth (i.e., the golden prediction). Let score $= 1$ if the output of $R$ is exactly the same as the ground-truth; otherwise, score $= 0$.

- **PO with Embedding Similarity**: scoring by soft-matching — Base LLM (GPT-3.5) with PO considering the cosine similarity score between the embedding vectors of the output and the ground-truth. The outputs are embedded by the powerful text-embedding-ada-002 model (OpenAI, 2024) via OpenAI APIs.

- **PLHF**: our proposed framework — the base LLMs $M_R$ and $M_E$ are both set to be raw GPT-3.5 (gpt-3.5-turbo-0125) models.

### 4.3 EVALUATIONS

For the model output, we employ multiple human experts as the judges to provide professional scores with respect to the score scales of the original data. However, for the public datasets, the original people who labeled the data are unavailable to give their judgement for our new generated outputs for the data inputs. In our experiments, we introduce a concept of *pseudo-human* judge to execute output evaluations. Specifically, we use GPT-4o with prompt optimizations via DSPy as the pseudo-human judge. Since we consider GPT-3.5 for the base LLMs in all the methods for experiments, the pseudo-human judge (i.e., GPT-4o with DSPy PO based on training samples) is a more powerful model that can provide fair evaluations toward output quality.

### 4.4 EXPERIMENTAL RESULTS

Table 2 lists the experimental results of the evaluator task. As shown in the table, we can observe that the conventional methods (MLP/SVM with Text Embedding) seem struggled in predicting the labeled scores from the given embedded inputs. In contrast, the LLM-based methods performed significantly better on both public datasets and the industrial tests. A possible reason is that LLMs

Table 3: Summary of experimental results for the responder subtask across each dataset. For the industrial dataset SQL-QA, the overall Accuracy score are given by actual human experts in the company; for the public datasets, the scores from the pseudo-human judge are shown. The values for Base LLM are the actual scores, whereas for the other methods, relative improvements are shown in percentages. The best ones are marked in bold font.

| PO | Method | SGD | AES-ASAP | AES-2.0 | SQL-QA |
|---|---|---|---|---|---|
| | Base LLM (GPT-3.5) | 4.25 | 26.50 | 5.35 | 0.74 |
| DSPy | PO with GPT-4o | +1.18% | +3.70% | +0.75% | +5.41% |
| | PO with Exact Matching | - 4.00% | -10.87% | - 5.61% | 0.00% |
| | PO with Embedding Similarity | +1.65% | - 4.27% | +0.56% | +10.81% |
| | PLHF | **+6.59%** | **+8.45%** | **+2.62%** | **+18.92%** |
| TextGrad | PO with GPT-4o | +4.71% | +3.28% | +1.31% | +2.70% |
| | PO with Exact Matching | -10.59% | -15.92% | -10.84% | -18.92% |
| | PO with Embedding Similarity | +3.53% | - 0.64% | +0.93% | +2.70% |
| | PLHF | **+8.71%** | **+8.68%** | **+4.30%** | **+18.92%** |

might have superior fitting and understanding capabilities to handle text inputs. With prompt optimizations, both DSPy and TextGrad provided more effective prompt for more accurate evaluators.

As for the experimental results of the responder task, shown in Table 3, we consider both DSPy and TextGrad as the prompt optimization (PO) tool for each method in the comparisons. Overall, the results are relatively similar in same directions for each pair of the scores between the two PO selections. In summary, for all the four datasets, the proposed PLHF framework achieved the best performance in output quality, in terms of the metric for each task. Moreover, PLHF used GPT-3.5 as the evaluator's base LLM $M_E$ to achieve superior performance than 'PO with GPT-4o', which conducted prompting with a more powerful GPT-4o as the evaluator. For the other baselines, 'PO with GPT-4o' consistently outperformed 'Base LLM (raw GPT-3.5)'. Regarding the conventional matching-based scoring functions, both hard-matching (Exact Matching) and soft-matching (Embedding Similarity) produced outputs with worse quality.

## 4.5 PERFORMANCE ANALYSIS

In addition to the overall performance, we also analyze the robustness and the relationship between model effectiveness and the number of training samples involved in the prompt optimization process. As examples, Figure 4 demonstrates the performance curves for datasets SGD and AES-ASAP. Note that, to analyze the performance of a responder with $n$ data samples, we also use the evaluator optimized with the same $n$ samples. For the curves of evaluator $E$, we can observe that the RMSE loss raised for initial samples, then the RMSE value dropped significantly after few shots of data. For the curves toward responder $R$, the pattern is similar to the curves of $E$ (but in opposite way), the output quality score dropped for initial samples, while the score then bouncing back and achieving new highs with greater number of samples. Last but not least, as expected, the standard deviations lower along with the increasing number of training samples.

## 5 CONCLUSION

In this paper, we focused on prompt optimizations for LLMs with a limited amount of human feedback — a more practical and achievable approach for real-world applications. To address the challenges of no well-defined metrics and the scarce human resources, we introduced PLHF, a few-shot prompt learning with an evaluator module design to automatically score the outputs generated by LLMs. We performed extensive experiments with public datasets and a real industrial dataset to verify the effectiveness of PLHF. The experimental results demonstrated that PLHF outperforms existing methods across from simple string matching functions to even the latest publicly available LLMs as output evaluators in terms of the output quality. Overall, our proposed framework is practically effective especially for the scenarios when directly applying pre-trained general-purpose LLMs are not the best option. Our future work involves enhancing and deploying the proposed framework across diverse applications, particularly for tasks that utilize multi-modal data.

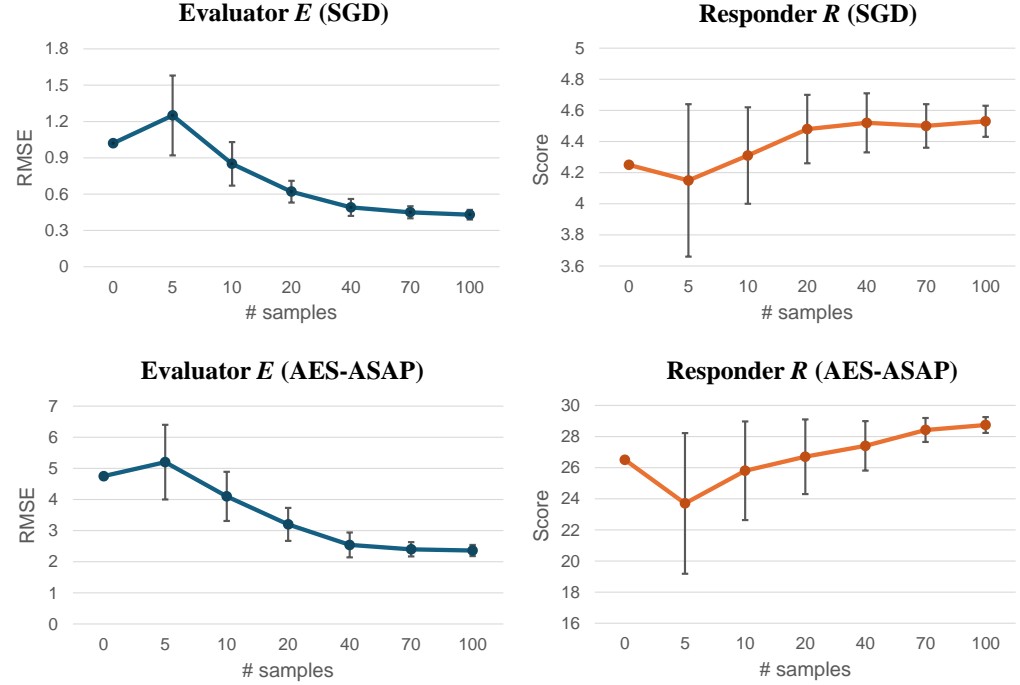

Figure 4: Performance curves of PLHF on the datasets SGD and AES-ASAP. For the plots, we consider DSPy as the PO method for PLHF. The $x$-values are the number of (randomly selected) training samples. The $y$-values are mean values of the RMSE losses for $E$ and the output scores for $R$, respectively. The vertical bar of each point indicates the standard deviations estimated in 30 runs.

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
