# OpenReview forum: "PLHF: Prompt Learning from Few-shot Human Feedback"
_ICLR.cc/2025/Conference — Submitted to ICLR 2025_

### Official Review · Reviewer_CC2k · 2024-10-27

**Soundness:** 2
**Presentation:** 2
**Contribution:** 2
**Rating:** 5
**Confidence:** 4

**Summary:**

This paper addresses the lack of effective metrics in prompt refinement by proposing the PLHF method, which leverages human feedback. The approach includes a responder and an evaluator, where the evaluator simulates human feedback scores to iteratively improve the responder’s output based on these scores. Experiments conducted on various datasets and tasks demonstrate the method's effectiveness.

**Strengths:**

1.The paper's motivation is sound. Using LLMs for scoring may be unreliable and unstable. This paper proposed PLHF adopts a few-shot approach to align LLM scoring by using human-scoring examples, which is both intuitive and reasonable in my view.

2. The iterative, minimal human feedback mechanism effectively enhances the quality of optimized prompts.

**Weaknesses:**

1.My main concern is whether using few-shot examples of human scoring as feedback for LLM prompt optimization can consistently work. Can it genuinely mitigate the inherent issue of relying on a generative model (LLM) that outputs discriminative scores based on next-token prediction? I believe providing human scoring examples may be partially effective as guidance during the LLM scoring process. However, I remain skeptical about the overall effectiveness of this approach in addressing the core problem.

2.This approach raises a significant issue: the diversity, quality, and downstream task coverage of the human scoring examples become critical. These factors could greatly influence the effectiveness of prompt optimization on unseen cases and introduce specific requirements for data collection.

3.Additionally, the writing makes it somewhat challenging to grasp the main focus. For example, the methodology section lacks formal descriptions, relying heavily on textual explanations that complicate the reader’s understanding. Furthermore, Algorithm 1 appears overly lengthy and complex, covering both training and testing aspects.

4.My final concern lies with the experimental section. The experiments are relatively weak, lacking benchmark results. It would be valuable to see how the proposed PLHF method performs on datasets like Vicuna Eval and Self-instruct Eval.

5.Additionally, there are closely related works [1] that could be discussed.
[1] Black-Box Prompt Optimization: Aligning Large Language Models without Model Training”

**Questions:**

Please refer to the weakness.

---

> ### Author Response · Authors · 2024-12-03
>
> Thank you for the thorough review and the constructive feedback toward our paper. For the mentioned concerns, please see the following responses.
>
> 1. For the addressed issue of whether an LLM can produce discriminative scores based on next-token prediction, we acknowledge that our paper might not be a universal solution to the core problem of LLMs. However, the main purpose of our work is to provide a framework that performs prompt optimization efficiently in terms of the number of human feedback calls, when there is no explicit well-defined metric for the target task. Overall, though we do not have theoretical proof toward the model effectiveness (at least for now), we still want to share our findings as a new possibility to solve the problem in this paper.
>
> 2. The reviewer’s concern in the diversity, quality, and downstream task coverage of the human scoring is absolutely right. The effectiveness and accuracy of the finalized prompt depends on the quality of human feedback. However, as long as the scores (or say, labels) are provided by real humans, almost all prompt optimization methods suffer from the same human labeling quality issue, because the prompting itself is fundamentally based on labeled observed/training samples. Since the proposed method is a framework for prompt optimizations, we would like to address that the mentioned concern is considered out-of-scope in our work.
>
> 3. We greatly appreciate your feedback towards the readability of our manuscript. We will attempt to improve the descriptions and organizations for the mentioned parts.
>
> 4 & 5. We agree that the Experiments section has a room for improvement. We indeed plan to add more baselines, comparisons and analyses to justify the effectiveness of our prompt optimization framework. Due to the problem setting, it might not so suitable to include the mentioned Vicuna Eval and Self-instruct Eval datasets into the comparisons. For this part, we will make it clearer in the later versions.
>
> Again, we greatly appreciate your review!

---

### Official Review · Reviewer_gwUD · 2024-10-31

**Soundness:** 3
**Presentation:** 2
**Contribution:** 2
**Rating:** 3
**Confidence:** 3

**Summary:**

This work proposed a few-shot prompt optimization framework that employs an evaluator module acting as a metric to estimate the output quality.  The framework requires only a single round of human feedback to complete the entire prompt optimization process.

**Strengths:**

Previous work relies on human feedback, whereas this study employs a prompt-optimized LLM as an evaluator to assess the output of LLM responses. By substituting human experts with an LLM, this approach enhances the automation of the evaluation process.

**Weaknesses:**

This work appears to be more focused on engineering applications rather than theoretical depth.
It is more suited for conferences like EMNLP, NAACL, or other NLP venues.
The contributions seem insufficient for an ICLR submission.

**Questions:**

1. the symbols are not clear, for instance, in Figure 2, the abbreviations appear confusing and difficult to read.
2. how to define the score threshold in Figure 3 is not clear to me.
3. How reliable are the training samples labeled by humans, is it possible humans have biases on the scores?
4. I am curious if there is an experiment on page 8 that utilizes the PLHF framework with the base LLMs configured as GPT-4.

---

> ### Author Response · Authors · 2024-12-03
>
> Thank you for the review. We would like to address the mentioned issues as follows.
>
> 1 & 2. We appreciate your feedback toward the paper readability. We will improve the notations and descriptions.
>
> 3. We would like to clarify --- the tackled task in this paper is in fact providing output evaluations (i.e., the scores) acting as from the real humans (since in the mentioned cases, there is no explicit well-defined metric for prompt optimizations). Therefore, whether the humans have biases on the scores is not considered as a concern in our work. After all, the main purpose of our framework is to perform prompt optimization to fit the human preference pattern, based on human feedback, so we assume that the human feedback is correct.
>
> 4. Since we adopt GPT-4o as the pseudo-human judge (described in Sec 4.3) to provide evaluations, we tend to adopt a weaker LLM (GPT-3.5 in our case) as the base LLM --- to imitate the relationship of LLMs and real human, where human experts should be more authoritative than LLMs. Nevertheless, we agree that we should consider other LLMs as the base LLM in our experiments.

---

### Official Review · Reviewer_wvyU · 2024-11-01

**Soundness:** 2
**Presentation:** 2
**Contribution:** 2
**Rating:** 3
**Confidence:** 4

**Summary:**

The paper introduces Prompt Learning with Human Feedback (PLHF), a framework designed to optimize the prompts of LLMs with limited human feedback. PLHF comprises two key components: a Responder module that generates outputs and an Evaluator module that estimates output quality based on human feedback. It starts by having human experts score a set of training samples, then optimizes the Evaluator module (i.e. update the Evaluator prompt) to mimic these scores. The Responder module is subsequently optimized (i.e. update the Responder prompt) to generate outputs that align with the Evaluator's scoring. Empirical results on public and industrial datasets demonstrate PLHF's superiority over existing output scoring strategies for LLM prompt optimizations.

**Strengths:**

- The paper is clear, technically sound, and presents a new framework (PLHF) for prompt optimization.
- The experimental results demonstrate the effectiveness of PLHF across various datasets.

**Weaknesses:**

- The paper introduces an automatic prompt optimization method. However, upon reviewing the experimental section, it was observed that the method requires task-specific training to obtain the task-specific optimized prompt, suggesting it is not universally applicable (training one prompt for all tasks). As depicted in Figure 3, the initial prompt appears to be quite simplistic. This raises the question of whether the performance could be significantly improved by manually crafting more complex initial prompts. The manual design cost might be less compared to the overhead of training for each task individually. For instance, in the math problem example on the right side of Figure 1, where user feedback indicates the problem is too easy, an obvious solution would be to specify the difficulty level in the initial prompt. It would be insightful to understand if the proposed framework still offers significant improvements when starting with a more complex initial prompt, as is common in practical scenarios.
- Regarding the prompt optimizations for the Responder, it is unclear if the goal is to maximize the score of the Evaluator. The rationale for using only positive samples for training is not explicitly stated. What would be the impact if all labeled samples were used for training instead? Additionally, on lines 247, it is mentioned that a manual score threshold is required, which seems to imply that it needs to be adjusted on a per-task basis.
- The experimental section lacks details on how the training and test sets were divided, and the term "few-shot" is used without specifying the exact number of shots.
- The number of rounds of experiments conducted for the results presented in Table 2 and 3 is not specified. Additionally, it is unclear if any significance testing was performed. As a reviewer, I cannot determine if the improvements obtained in Table 3 are statistically significant, especially considering that the scores are derived from a somewhat stochastic GPT model.
- The paper would benefit from a more detailed discussion on the limitations of the PLHF framework and potential directions for future research in the conclusion section.

**Questions:**

- What is the impact of more complex initial prompt design on the performance of the proposed framework?
- How does the manual score threshold affect the outcomes?
- Can the authors clarify how the training and test sets were divided for the SGD and AES datasets? This is particularly crucial given the emphasis on the few-shot setting.
- Were significance tests conducted for the results in Table 2 and 3? it would be beneficial for the authors to report the variance of results, especially considering that the test results are derived from the somewhat stochastic nature of GPT models?
- Table 3 shows a notably poor performance of the PO with Exact Matching method. Could the authors provide insights into why this method performed so poorly compared to other baselines?
- On line 320, the term "domain 1" is mentioned, but it is unclear what this refers to within the context of the paper.

Typos:
- On line 083, the paper mentions "PO" which seems to be repeated.

---

> ### Author Response · Authors · 2024-12-03
>
> We greatly appreciate your constructive review. Each mentioned point from the reviewer definitely makes us realize that our manuscript could be refined with more enhancements. For our answers to the questions, we summarize as follows.
>
> - We will further expand the experiments to include more analyses toward model effectiveness and robustness.
> - In a new version of our paper, the details of dataset pre-processing will be provided in the appendix section.
> - More insights and examples are also planned to be presented in the appendix section.
> - For the term “domain 1” mentioned on Line 320, it is originally from the dataset description of the AES-ASAP dataset (Ben et al. 2012). Since some of the essays have scores in two different domains, we specify that we consider the human-labeled score in domain 1. We will make the related descriptions more self-contained to avoid unnecessary confusion in the future version of our manuscript.

---

### Official Review · Reviewer_DefM · 2024-11-03

**Soundness:** 3
**Presentation:** 3
**Contribution:** 3
**Rating:** 6
**Confidence:** 4

**Summary:**

The paper studies the problem of prompt optimization which has recently gained significant attention in the community. In particular, the key focus of the paper is prompt optimization in situations when 1) no automated evaluation metrics for scoring generated outputs is available (as in MMLU, MATH, GSM8K etc) and 2) an automated evaluation / scoring using existing LLMs (e.g., GPT4o) are not reliable w.r.t to human evaluations (Fig.1). To address this, the authors explore a new approach referred to as PLHF, which aims to perform prompt optimization while using atmost linear number of human annotations w.r.t to the underlying dataset. The proposed approach consists of two main modules: responder LLM R and evaluator LLM E. The core of the approach boils down to optimizing the evaluator LLM E using few-shot training samples and then using the same to obtain the optimized prompt P for the responder module R (Alg.1). Quantitative experiments are provided on the three subjective evaluation datasets (e.g., Automated essay scoring) in order to demonstrate the efficacy of the proposed approach.

**Strengths:**

* The paper studies an important problem of performing prompt optimization for subjective tasks where 1) no objective evaluation is possible, and 2) automated scoring using existing LLMs is not feasible.
* The authors demonstrate consistent improvements across diverse tasks over prior works for prompt optimization when using limited samples and evaluating on subjective tasks
* The paper consists of good figures and examples which highlight the problem with using automated LLM metrics for evaluation.

**Weaknesses:**

* One of my main concerns is limiting to gpt3.5 as the base model for the results presented in the paper.
      - While the paper shows prompt optimization results with GPT4o, the same also use GPT4o as the evaluator alone while still using GPT3.5 as the base model
      - Therefore, its not clear performance improvements from the proposed approach are limited to weaker models, or can they be extended to stronger models as well such as GPT4o, LLaMA-3.2 etc.
* Also in terms of the technical contribution, it seems that the proposed approach boils down to performing an additional prompt optimization w.r.t to the evaluator model before using the same for optimizing the prompt using TextGrad or DSPy.
     - For instance, instead of asking GPT4o (evaluator LLM) for rating the prompt, the proposed approach first provides additional prompt optimization for the evaluator prompt before then using the same for prompt optimization.
    - If so, then it would be more beneficial and clearer to state the same upfront in order to put the novelty of the paper in a more clear fashion.
   - Also while Fig.~3 seems to contain examples of initial and optimized prompts for the evaluator, it seems the optimized prompts largely consist of the in-context examples from the human annotations. If so, how is this different from simply providing the human annotations as additional context for the evaluator and responder model
 - It would be more useful to provide detailed initial and optimized prompts (similar to OPRO paper), demonstrating the what the optimization results look like.
* Finally, it would be beneficial to have a plot of the observed performance of the base model on different tasks, while varying the number of samples.
       - While Fig.~4 contains the plot of responder and evaluator LLM performance w.r.t to number of samples, it does not contain a plot for the final performance of the proposed approach as well as the baselines.

**Questions:**

Please refer the weakness section above.

---

> ### Author Response · Authors · 2024-11-27
>
> Thank you for your recognition of our work and for your insightful reviews. For the mentioned weaknesses, our responses are as follows.
> 1. We agree with the point that we should include various LLMs to act as the base LLM in our experiments. However, since we consider GPT-4o as the pseudo-human judge to make overall evaluations, we tend to select a weaker LLM (GPT-3.5 in our case) as the base LLM (to mimic the relationship of LLMs and real human, in our target scenarios, human experts should be more accurate than LLMs). But yes, we will add more alternatives to replace the GPT-3.5 and GPT-4o, respectively, in our future experiments.
> 2. We appreciate the suggestions of making our novelty clearer. For the mentioned question, we would like to explain that we only require human annotations for the score of each training sample. The idea is that scoring an input-output pair (for the evaluator E) is relatively easier than providing suggested output for a given input (for the responder R). But we agree with the reviewer, providing the human annotations as additional context for both E and R is also a possible way to perform prompt optimizations with human feedback.
> 3. Yes, we sincerely accept the idea of including detailed the initial prompts and the corresponding optimized prompts for each task in our experiments. We have started to prepare it and we expect such results will be available as the appendix of our paper in the upcoming camera-ready version.
> 4. We also greatly appreciate the suggestion of adding the plots showing the performance changes with various number of training samples in the analysis part for the baseline models. Extra plots will enhance the clarity of our existing comparisons between the proposed framework and the baseline combinations.

---

### Official Review · Reviewer_DNc4 · 2024-11-04

**Soundness:** 2
**Presentation:** 2
**Contribution:** 2
**Rating:** 5
**Confidence:** 4

**Summary:**

This paper proposes a method for few-shot prompt optimization using a small amount of human feedback, aiming to address issues in scenarios where there are no well-established evaluation metrics. Specifically, the authors decompose the system into two modules: the evaluator and the responser, and perform interactive optimization between the two modules. The authors compare their approach with mainstream baseline models on both public datasets and industrial datasets.

**Strengths:**

1. The paper is well written and easy to follow;
2. This paper investigates prompt optimization in scenarios without a clear metric, which is a very important research question;
3. The idea of introducing human feedback into prompt optimization is valuable.

**Weaknesses:**

1. As shown in Figure 3, the prompt optimization for E and R appears to only add a few few-shot examples, which is implemented based on methods like DSPy. This form of optimization that focuses solely on few-shot examples is relatively narrow, and the author needs to conduct a more comprehensive comparison with other prompt optimization methods.
2. The experimental baselines compared in the paper do not incorporate human annotations and feedback. The author should compare with methods that also introduce human feedback, such as Prompt Optimization with Human Feedback.
3. The modeling for the iterative optimization of E and R is relatively simple, involving first optimizing E and then optimizing R based on the guidance from E. The author needs to compare this with other iterative optimization methods and provide some theoretical analysis to support it.
4. Most of the experiments in the paper were conducted using GPT-3.5, and additional experiments with other models are needed to verify the generalizability.

**Questions:**

Please see the weaknesses.

---

> ### Author Response · Authors · 2024-11-27
>
> We are sincerely grateful for the reviews toward our work. For the concerned issues:
> 1. As we mentioned throughout the entire paper, the proposed framework is especially designed for few-shot prompt optimization. While we acknowledge the idea of comparing our model with a wider range of prompt optimization methods, the original setting of the task is that only few shots of samples are available for prompt optimization. We will improve our writing to mitigate the ambiguity on the scope.
> 2. Prompt Optimization with Human Feedback is indeed a related work providing another strategy to leverage human feedback to perform prompt optimization for LLMs. However, since the source paper is not officially published yet when we made this submission (the manuscript of “Prompt Optimization with Human Feedback” is also submitted to ICLR 2025 this time), based on the convention, we believe that it is reasonable to not include such work in our submission.
> 3. To our best knowledge, with the constraint of few-shot samples and no explicitly available metric to evaluate the LLMs’ outputs, there is no existing iterative method to perform prompt optimizations. For the theoretical analysis, we will consider it as future work. Similar to other work developing LLM usages, empirical results are often prior than the theoretical proof for a new concept. The main purpose of our paper is to demonstrate a novel possibility to tackle the mentioned issues.
> 4. Indeed, agreed with the reviewer, we are considering to include other LLMs are the base LLM in our experiments to enhance the coverage of our experiments.

---

### Comment · Area_Chair_SQ7C · 2024-12-04
**The deadline of the discussion period (12/3) is approaching**

Dear Reviewers,

      The authors have posted their response to your concerns. If you have not yet checked them, please take a look and provide your further feedback to see if your concerns are well-addressed at your earliest convenience. Thank you very much.

Best,
AC

---

### Meta-Review · Area_Chair_SQ7C · 2024-12-24

**Metareview:**

This work introduces a prompt optimization framework for large language models (LLMs) that addresses the lack of a clear evaluation metric by leveraging a prompt-optimized LLM as the evaluator, guided by few-shot human feedback. The proposed method, PLHF, demonstrates superior performance on both public and industrial datasets compared to existing scoring strategies for LLM prompt optimization. Most reviewers agree that the paper is well-written and easy to follow, and they acknowledge the significance of addressing the challenge posed by the absence of a clear metric for LLM prompt optimization. However, reviewers noted that the paper lacks sufficient evaluations of the proposed method on the latest LLMs beyond GPT-3.5, across various tasks and benchmarks (DNc4, DefM, CC2k, gwUD). Additionally, the comparisons with other prompt optimization techniques that utilize human feedback or iterative optimization (DNc4) are limited.  They also raised the concerns about the influence of initial prompt design on the final optimized prompt (DefM, wvyU). Some hyperparameter descriptions, such as the score threshold, were also noted to be unclear (gwUD, wvyU). As a result, reviewers unanimously providing negative feedback, and the paper received an average rating of 4.4 finally. The authors are encouraged to address the reviewers' comments and refine the paper for resubmission to a future venue.

**Additional Comments On Reviewer Discussion:**

The reviewers expressed concerns about the paper’s limited evaluation of the proposed method on state-of-the-art LLMs beyond GPT-3.5, across diverse tasks and benchmarks (DNc4, DefM, CC2k, gwUD). They also noted a lack of comprehensive comparisons with other prompt optimization techniques that incorporate human feedback or iterative optimization (DNc4). Additionally, questions were raised about the impact of initial prompt design on the optimized prompt (DefM, wvyU) and the unclear description of certain hyperparameters, such as the score threshold (gwUD, wvyU). Despite these concerns, the authors did not provide clear responses or update the paper with additional experimental results during the rebuttal period.

---

### Decision · Program_Chairs · 2025-01-22

Reject